# Shaping Antitumor Immunity with Peptide Vaccines: Implications of Immune Modulation at the Vaccine Site

**DOI:** 10.3390/vaccines13111150

**Published:** 2025-11-11

**Authors:** Amrita Sarkar, Emily Pauline Rabinovich, Craig Lee Slingluff

**Affiliations:** 1Department of Surgery, Division of Surgical Oncology and the Human Immune Therapy Center, Cancer Center, University of Virginia Health System, Charlottesville, VA 22903, USA; 2School of Medicine, Cancer Center, University of Virginia, Charlottesville, VA 22903, USA

**Keywords:** peptide vaccines, cancer vaccines, antitumor immunity, vaccine site, tumor immunology, immune modulation

## Abstract

Cancer vaccines have emerged as a class of therapeutics designed to harness the immune system to stimulate durable anti-tumor responses with lower systemic toxicity than conventional therapies. Many platforms have been explored, including protein, peptide, DNA, RNA, and cell-based vaccines. Within this landscape, peptide vaccines remain a promising approach. Most clinical trials have examined peripheral immune responses and clinical outcomes, but there is growing interest in the vaccine site microenvironment (VSME) as a window to understand local immune activation and its implications for systemic immunity and tumor control. Studies of the VSME have investigated the effects of adjuvants, local immune cell dynamics, and their correlation with systemic responses and outcomes. Local adjuvants typically enhance immune cell infiltration, though there are concerns regarding VSME sequestration or dysfunction of immune cells, which could impact systemic efficacy. Repeated vaccination at a single site may improve antigen presentation and immune responses, but factors such as injection site location may be linked to variability in clinical outcomes. Current studies are limited by substantial variability in sampling, timing, and analyses used in VSME assessment. This limits the comparability of findings and broader inferences regarding the influence of vaccine site dynamics on therapeutic efficacy. Standardized VSME assessment as part of future vaccine trials may improve evaluation of immune responses and provide a more consistent surrogate for vaccine effectiveness. This refinement may inform optimal vaccine strategies and further support the development of next-generation cancer immunotherapies.

## 1. Introduction

In the United States, cancer is the second leading cause of death; in patients under the age of 85, it remains the leading cause of death overall [1]. Despite advances in cancer screening, early diagnosis, and treatment, aspects of cancer biology, including immune evasion and immunosuppression in the tumor microenvironment (TME), continue to pose barriers to developing efficacious and specific anti-tumor therapies [2]. Immune checkpoint inhibitors have been approved for the treatment of various cancers. These medications are designed to block inhibitory signaling that may prevent T-cell-mediated anti-tumor immunity. Despite this, various malignancies continue to demonstrate either de novo or adaptive immunity to these immunotherapeutics [3]. Additionally, the use of checkpoint inhibitors and other existing anti-tumor drugs can be limited by their risk profiles, which include serious immune-related adverse events such as autoimmune myocarditis, myasthenia gravis, and pneumonitis, among others [4].

The clinical benefits of checkpoint therapy have further highlighted the potential for a patient’s immune system to mount an effective response against tumor cells and to enable immune control in patients with primary or secondary resistance to checkpoint blockade. Vaccine development for cancer has focused on optimizing delivery and subsequent immune responses against both tumor-associated and tumor-specific antigens. Various strategies have been explored in designing cancer vaccines, including DNA, RNA, protein, peptide, and cell-based vaccines [5]. Messenger RNA (mRNA) based vaccines have been explored extensively as cancer therapeutics; they offer relative ease of protein expression compared to DNA vaccines and lower risk of incorporation into the host genome [6]. These vaccines do not contain an infectious component and are typically well tolerated, with rapid degradation of vaccine components reducing risk of long-term adverse effects [7]. Additionally, mRNA vaccines are inherently immunogenic, stimulating both innate and adaptive immune responses and minimizing the need for additional adjuvants [8,9]. Early challenges in mRNA vaccine development included the inherent instability of these antigens, which have since been addressed through molecular engineering approaches designed to enhance stability, delivery, and uptake [6,7]. Although mRNA vaccines are attractive for their capacity to enable personalized vaccines based on individual tumor neoantigens, peptide-based vaccines may offer additional stability and implementation across diverse clinical settings [10]. Tumor cell-based vaccines can utilize whole tumor cells, often irradiated to reduce disease risk, to produce anti-tumor immune responses [11]. Alternatively, dendritic cell (DC) vaccines may use dendritic cells matured and stimulated in vitro to overcome tumor microenvironment immunosuppression, which may ordinarily limit DC function [11,12]. Viral vectors have also been explored as mechanisms to express tumor antigens or infect antigen-presenting cells (APCs) to express tumor antigen transgenes. Although these vectors are often inherently immunogenic, vector-specific challenges must be considered in vaccine design, including host immunity to the viral vector and potential for neurotropism and infectious complications [13]. Vaccine types with lower inherent immunogenicity may require administration with local adjuvants to increase vaccine response and tumor infiltration by immunologically active cells [14].

Peptide vaccines are subunit vaccines utilizing epitopes to directly stimulate an immune response against tumor cells. The earliest peptide vaccines utilized proteins that were unmutated but overexpressed in tumors compared to normal human tissues [15]. These vaccines often aim to stimulate CD8^+^ T cell responses via class I MHC-restricted epitopes but may also contain helper peptides to simultaneously induce a CD4^+^ T cell response [16]. As an example, Mucin 1 (MUC1) is a clinically significant tumor-associated antigen in breast cancer vaccine development, as it is overexpressed in most breast cancers and absent in normal tissue [17]. The ABCSF 34 trial in breast cancer vaccine development studied a MUC1-based vaccine, tecemotide, that in combination with neoadjuvant systemic therapy for early breast cancer significantly improved recurrence-free and overall survival [18]. Newer generation vaccines have also targeted both mutated epitopes and aberrantly expressed non-mutated self-antigens identified on cancer cells. Additional epitopes in cancer vaccines may also include frameshift-derived products, single nucleotide polymorphism-associated variants, cancer-testis and oncofetal antigens, and lineage-specific or differentiation antigens, such as melanocytic antigens [19,20]. Peptide vaccines are generally well-tolerated, with the most common adverse effects including local pain, swelling, erythema, and fever [21,22,23]. Most trials have reported low rates or absence of grade III/IV toxicities [24]. However, these vaccines have demonstrated variability in their clinical efficacy; some have proven more effective than others in producing positive, durable systemic immune responses to the included epitopes. Additionally, some trials have demonstrated low or variable correlation between the generation of systemic immune response and tumor regression in vaccinated participants [25]. The variability in response to these early peptide vaccines has been attributed to tumor evasion of immune surveillance via alteration of antigen expression, barriers to immune cell infiltration into the TME, or immunosuppression in the TME. As a result, some recent studies have focused on combining these peptides with adjuvants, immunotherapeutic drugs, and helper peptides to broaden and strengthen the immune response, with the ultimate goal of improving overall clinical outcomes [24]. Additionally, central tolerance may limit immune responses to tumor-associated antigens (TAAs), leading to a shift in focus toward tumor-specific antigens (TSAs), including neoantigens, that are not expressed on normal cells.

While vaccines have been traditionally viewed as a means to deliver antigen to draining lymph nodes for further processing and T cell priming [26], accumulating evidence suggests that important immunologic events may also be occurring directly within the vaccine site microenvironment (VSME). This review highlights the VSME as an emerging focus of study in vaccine development and testing. The authors discuss known elements of this underexplored niche where adjuvant effects, antigen presentation, and immune cell responses converge to influence downstream anti-tumor immunity.

## 2. The Vaccine Site: An Initial Interface for Antigen Presentation

Following administration of a vaccine, the vaccine site serves as the first point of contact between antigens and elements of the patient’s innate and adaptive immune systems. This establishes a local immune microenvironment that may shape antigen uptake and processing, impact downstream responses, and build the foundations of a subsequent systemic immune response.

At the site of injection, vaccine components result in the trafficking of innate and adaptive immune cells. The nature of early inflammatory reactions and the recruitment of APCs, including DCs and macrophages, result in cytokine and chemokine signature changes. Previous work by Clancy-Thompson et al. examining the VSME following administration of a peptide vaccine demonstrated increases in levels of IFN-γ, and IL-12 [27]. Similarly, studies of vaccine site changes following administration of mRNA vaccines have identified upregulation of numerous pro-inflammatory genes [28] with infiltration by neutrophils, macrophages, lymphocytes [29], and a variety of monocyte subtypes [28].

Initial safety analysis of a melanoma peptide vaccine by Hu et al. demonstrated significant correlation between vaccine site inflammatory adverse events and both development of immune response and subsequent disease-free survival; this further suggests that the initial inflammatory response to the vaccine may be fundamentally influential on clinical outcome [30].

## 3. Adjuvant-Mediated Recruitment and Activation of Immune Cells

Adjuvants are utilized to elicit a local inflammatory response, to enhance antigen presentation, and subsequently to improve recruitment of APCs, T cells, and B cells to the vaccine site. These early signals may govern the magnitude and quality of T cell responses to the chosen antigens.

Inclusion of adjuvants is associated with chemokine secretion and recruitment of various immune cells to the vaccine site. Aluminum adjuvants have been widely utilized as vaccine adjuvants, in both humans and animals, as stimulators of the Th2 response for antibody responses [31]. These compounds form insoluble particles to stimulate DC [32] and macrophage uptake, resulting in inflammasome activation and production of pro-inflammatory cytokines [33]. A notable limitation of these adjuvants with regard to their use in cancer vaccines is their poor induction of Th1 response and cellular immunity. These limitations have been addressed through administration with additional adjuvants [34] or through bioengineering techniques including mesostrand [35] and nanoparticle [36] creation intended to augment Th1 responses.

Some adjuvants create an antigen depot at the vaccine site to prolong exposure to APCs [37]. Early evidence supporting this included significant increases in antibody responses to injected ovalbumin with addition of mineral-oil adjuvant. This increased response was noted to persist for over a year after injection, significantly longer than injection without adjuvant. Emulsions isolated from mice months after injection demonstrated a stratified appearance with an oil layer separated from the emulsion layer, supporting the hypothesis that prolonged breakdown of the emulsion allows depot formation with slow antigen release over time [38]. A study of early inflammatory changes following intradermal injection of incomplete Freund’s adjuvant (IFA), a water-in-oil emulsion, demonstrated significant increases in TNF-α and IFN-γ mRNA in draining inguinal lymph nodes. With the addition of type II collagen to IFA, levels of TNF-α, IL-2, and IFN-γ were increased; harvested lymph node cells also produced a strong TNF-α mRNA response following in vitro re-stimulation with type II collagen [39]. Examination of mouse splenocytes following intraperitoneal administration of IFA has demonstrated increases in IL-1, 4, 5, 6, and 13 mRNA-producing cells; these chemokine changes may contribute to initiation of the immune response, Th2 stimulation, class switching, B cell proliferation, and antibody production [40]. Additionally, inflammatory responses to oil-based adjuvants may involve generation of oxygen radicals, as they are attenuated by catalase and antioxidant administration [41].

New developments in adjuvant strategies have demonstrated promise through their impacts on the VSME. For example, CpG oligodeoxynucleotides are able to induce maturation of plasmacytoid DCs into potent APCs, which subsequently secrete chemokines, causing lymphocytic migration to the vaccine site [42]. An examination of a novel IL-2/anti-IL-2 antibody complex adjuvant demonstrated significant increases in populations of DCs, CD8^+^ T cells, and natural killer cells at the vaccine site, in addition to a significant fold increase in circulating antigen-specific T cells when compared to vaccine alone [43]. This may be a promising new adjuvant to augment responses to peptide vaccines. Unique advances may also allow optimization of antigen presentation at the vaccine site to boost the subsequent immune response. For example, the creation of a stable gel matrix from poly-N-acetyl-glucosamine has allowed extended release of peptide and GM-CSF following injection; sustained release has been associated with locally increased DC activation, immune infiltration, and increased antigen-specific T cells in both the circulation and draining lymph nodes [44]. Collectively, investigations into diverse adjuvants have yielded heterogeneous results with respect to efficacy and effects on both the VSME and systemic immune responses (Table 1), underscoring the need for further systematic investigation.
vaccines-13-01150-t001_Table 1Table 1Comparison of mechanisms of action, advantages, and disadvantages of adjuvants used with peptide-based cancer vaccines.AdjuvantMechanismAdvantagesDisadvantagesMineral saltsDepotTh2 activationSafe, well-tolerated [31]Novel engineering techniques allow improved Th1 activation [35,36]May be utilized with additional adjuvants to augment response [34]Primarily Th2 activators with suboptimal activation of Th1 response and cellular immunityWater-in-oil emulsionsDepotSafe, generally well-toleratedAugmentation of immunogenicity and T cell responses [45,46]Evidence of activated T cell recruitment to VSME [47]Local adverse inflammatory effectsEvidence of T cell sequestration and dysfunction at vaccine site [48]TLR agonistsImmunopotentiation  CpG oligodeoxynucleotides Induce DC maturation with subsequent lymphocytic migration to VSME [42]Limited antigen-specific adjuvant effect [42]Poly-ICLC Enhanced DC activation, maturation, cross-presentation [49]Increased infiltration of CD8^+^ T cells into tumor [50]Transient effects on VSME cellular changes [49]May have increased effectiveness with systemic administration [50]TLR2 agonist Successful maturation of DCs and activation of APCs [51,52]Enhancement of Th1 activation [51]Tumor-reactive T cell responses need further investigation [53]TLR1/2 agonist Induction of CD8^+^ and Th1 responses in human [54]Effector memory T cell accumulation at injection site [54]Immune response demonstrated in one human subject [54]Phase I trials in process [55]SaponinsImmunopotentiation  QS-21 (AS15) Induction of CD8^+^ T cell responses [56]Increased antibody production and cellular response after peptide vaccination [57]Effective with intradermal administration [58]Local inflammatory and grade 1–2 systemic adverse effects [57,58]No clinically significant effect compared to placebo in phase III trial [59]IL-2/anti-IL-2 complexImmunopotentiationDrives DC expansion at vaccine site, draining lymph nodes, spleen [43]Increases CD8^+^ T cell and NK cells at vaccine site [43]Assessed in combination with mesoporous silica rod vaccine [43]GM-CSFImmunopotentiationGenerally safe and tolerated [60]Some studies demonstrating improved immune response [61]Some local inflammatory and systemic adverse effects [60]Randomized trials show decreased T cell responses or no impact, when added to other effective adjuvants [60,62,63]Stabilizing gel matricesDepotSustained cytokine release at vaccine siteEnhanced inflammatory infiltrate and antigen-specific immunity [44]Early phase development [44]

## 4. Antigen-Specific T Cell Infiltration, Retention, and Potential Dysfunction at the Vaccine Site

Once primed, vaccine-induced T cells infiltrate the site of antigen deposition, where they may persist, undergo functional modulation, or enter the systemic circulation to target distant sites (Figure 1). Retention of these activated T cells can potentially amplify local immune activity, but murine studies and clinical trials have suggested conflicting theories regarding the impact of the VSME on T cell function after activation.

One area of focus in vaccine site studies has been the examination of changes in antigen-specific and innate immune cell populations following the administration of peptide vaccines, with and without adjuvants. Roberti et al. identified vaccine-specific CD8^+^ T cell receptors (TCR) from a skin biopsy following administration of a neoantigen vaccine with IFA; this highlights not only the efficacy of this vaccine, but also emphasizes the utility of the vaccine site in reflecting a vaccine-specific immune response [64].

Some concerns have been raised in murine models that vaccination with a minimal epitope melanoma peptide in IFA led to local inflammatory changes that recruited tumor-specific CD8^+^ T cells to the vaccine site instead of to the tumor itself. Furthermore, antigen-specific T cells died in the VSME [48]. These findings have raised concerns about use of IFA with peptide vaccines; however, those murine studies identified this concern only with a 9-mer (minimal epitope) peptide, but this was not replicated when a longer peptide was used in the vaccine. In contrast, vaccine formulations that led to less antigen persistence at the site (CD40 antibody plus TLR agonist) shifted T cell localization toward tumors [48]. Additionally, investigation by Hailemichael et al. on the impact of adjuvant IFA with concomitant anti-CTLA-4 therapy demonstrated sequestration and destruction of anti-CTLA-4 induced effector T cells that were specific for non-vaccine tumor antigens [65]. In humans, vaccination with peptides in IFA also leads to T cell accumulation at vaccine sites [45]. One proposed mechanism for T cell retention in inflamed sites involves upregulation of integrins α1β1 (VLA-1) and αEβ7 (CD103) which are markers of tissue-resident memory T cells and enable adherence of T cells to collagen IV and E-cadherin in peripheral tissues; these have been identified at high frequency in CD8^+^ T cells infiltrating the VSME (after peptide + IFA vaccines) and in the melanoma tumor microenvironment [46,66,67] and may contribute to both the acquisition of a memory-like phenotype and enhanced lymphocytic trafficking and motility [68].

Despite concerns about T cell accumulation at vaccine sites, human clinical trials do support use of IFA as a vaccine adjuvant [69]. Clinical trials in humans have tested whether IFA depleted antigen-specific T cells in circulation when added to a minimal epitope peptide vaccine using peptide plus a TLR agonist. In two trials, addition of IFA to peptide vaccines has enhanced immunogenicity and T cell responses, measured by IFN-γ ELISpot assay, compared to vaccine administration with adjuvant TLR agonists alone [70,71]. The Mel58 trial [NCT01585350] demonstrated that IFA enhanced the magnitude and durability of CD8^+^ T cell responses to melanoma peptides when used in conjunction with TLR agonists rather than diminishing them [70].

Thus, human clinical trial data do not support the same concerns raised in the murine studies mentioned above. Despite the elegant design of those murine studies, they did not reflect the setting of human cancer vaccines—in those murine studies, T cells transgenic for the melanoma antigen were given adoptively, leading to dramatically greater frequencies of melanoma antigen-reactive T cells than found in melanoma patients. Also, the dose of IFA administered to the mouse, if translated to the human on a weight:weight basis, would be comparable to giving about 500 mL of IFA, rather than the typical dose of about 1 mL in most human clinical trials. Thus, the selective homing of T cells to the vaccine sites in that mouse model probably do not reflect the normal biology in human cancer patients.

To understand cellular and molecular effects of peptide vaccines and their adjuvants, our group has studied vaccine sites after 1 or more vaccinations with IFA or other adjuvants, and with or without melanoma peptide vaccine. We have identified a high proportion of activated, antigen-specific CD8^+^ T cells in the VSME, particularly when peptide was present [46]. There was evidence that those T cells in the VSME may be dysfunctional, likely explained by continuous antigen stimulation at the VSME [46]. On the other hand, analyses of human vaccine sites after use of IFA have revealed increased markers of mature dendritic cells; increased CD40 and CD40 ligand expression [47], enhanced Th1 signaling [47,49], and reduced arginase expression [47], especially after 3 vaccines at the same site. Collectively, current evidence indicates that IFA significantly shapes the VSME through inflammatory signaling, immune cell recruitment, and enhancement of downstream systemic immune responses relative to other adjuvants. Additional studies have further supported that adjuvants including TLR agonists (i.e., poly-ICLC and CpG) can support DC activation and T cell priming, either with TLR agonists alone or by adding TLR agonists to IFA [49,50,72]. However, many of the precise molecular and cellular mechanisms underlying IFA-induced alterations within the VSME remain incompletely understood, warranting further detailed investigation.

## 5. Evidence of Tertiary Lymphoid Architecture at the Vaccine Site

Histologic and molecular studies have suggested the emergence of tertiary lymphoid structures (TLS) within the VSME [46,49,73]. These ectopic lymphoid aggregates consist of a wide range of immunologically active cells, including T cells, B cells, and mature DCs. These cells surround high endothelial venules expressing peripheral node addressin (PNAd) [74,75]. TLS provide a scaffold for local antigen presentation and immune activation. Presence of TLS in tumor is linked to improved responses to immune checkpoint blockade therapy and clinical outcomes in a number of malignancies [76,77,78,79,80,81,82]. The addition of both IFA and QS-21 as adjuvants has induced formation of tertiary lymphoid structures at the vaccine site [45]. Current evidence suggests that depot-forming adjuvants function, in part, by enabling sustained antigen release [38,48] and continued presentation to immune cells within the VSME (Figure 2A). In both immature lymphoid aggregates and mature TLS, complex intracellular communication allows the presentation of vaccine antigens to immune cells, ultimately resulting in the production of effector and cytotoxic T cells, plasma cells, and memory B cells. A recent study of intratumoral immune signaling highlights immune triads as critical to tumor cell elimination; these triads are composed of CD4^+^ and CD8^+^ T cells engaging with the same dendritic cell, with the spatial positioning of these cells serving as a key predictor of anti-tumor activity [83]. Although not yet specifically studied in VSME TLS, these triads may also form in these immune structures and facilitate integral effector phase signaling (Figure 2B, step d). The evidence of TLS formation at the vaccine site further supports the VSME as an active center of immune organization. Further evaluation of TLS and related lymphoid structures within the VSME may provide important insights into the mechanisms driving vaccine-induced immunity.

## 6. Influence of Vaccine Strategies on Local Immune Signatures and Activity

The location and kinetics of vaccine delivery have the potential to significantly impact the immune contexture at the local VSME, shifting the balance of Th1/Th2 polarization, effector signatures, and regulatory pathways such as arginase activity. These vaccine site dynamics highlight the importance of studying vaccination strategies in determining both the quality and durability of subsequent immune responses.

A factor examined in relation to vaccine site dynamics is vaccine timing and location. Repeated vaccination with peptide and IFA at the same location induced immune signatures for accumulation of mature DCs and an increase in immature DCs over time. Additionally identified was a reversal of an initial Th2-dominant microenvironment, with a concurrent increase in CD4^+^ and CD8^+^ FoxP3^+^ cells and eosinophils after the third week of vaccination [84]. Meneveau et al. also studied the effects of IFA on the VSME, revealing increased expression of CD80, CD83, CD86, CD40, and CD40L with repeated vaccination at a single site. Additionally, single-site vaccination demonstrated a reduction in arginase-1, which can suppress T cell function; there was a concurrent increase in the expression of TLR adaptor molecules and chemokines associated with the formation of TLS [47]. Gene expression analyses (RNA sequencing) of VSME biopsies has revealed that peptide vaccines with IFA had improved antigen presentation pathways, and that repeated vaccination at a single site led to vaccine sites with more activated DCs, TLR adaptor expression, and Th1 dominance compared to a rotating site vaccination strategy [49].

## 7. Vaccine Strategies: Immunologic Consequences of Vaccine Site Selection

Existing literature also suggests that the local immune microenvironment may be influenced by vaccine site selection. Preclinical and clinical evidence have revealed that site selection affects antigen-specific antibody production, T cell modulation, and adjuvant activity, with significant implications for vaccine efficacy [49,85,86].

The Mel43 clinical trial [NCT00089193] focused on GM-CSF as a vaccine adjuvant and also noted a positive, but statistically insignificant, trend towards improved immunogenicity with repeated vaccination at two sites, as opposed to one site [60]. There may also be some importance as to the location of vaccine administration with regard to distance from the tumor of interest; an animal study of a polyICLC plus peptide vaccine for malignant glioma demonstrated decreases in antigen-specific TCRs, TCR affinity, effector function, and infiltration into the brain as the vaccine site moved closer to the tumor [87]. This suggests a need for possible spatial consideration of tumor-mediated immune suppression when planning vaccination strategies.

## 8. Translational Impact and Challenges in Peptide Vaccine Site Analysis

Existing literature focusing on peptide vaccines in melanoma has highlighted the importance of examination of the VSME as a valuable surrogate for understanding the local immunologic niche, which can both reflect and shape systemic anti-tumor responses. The aforementioned peptide vaccine trials provide a strong rationale for leveraging the use of vaccine site analysis as an early biomarker of vaccine efficacy and as a guide for future vaccine development. The VSME can offer insight into elements such as DC activation, helper T cell support, and cytotoxic T cell priming [45,47,49].

However, studies such as the aforementioned gp100 vaccine trials illustrate both the promise and limitations of peptide vaccination and evaluation of the vaccine site. Although the peptide vaccination combined with Montanide and high-dose IL-2 elicited detectable antigen-specific T cells locally, many of these cells displayed poor persistence and limited effector function [88,89]. This discordance between the evidence of local immune priming and lack of durable systemic efficacy underscores a critical translational challenge. While vaccine site evaluation has the potential to reveal whether an immune response is being generated locally, it may not reliably translate into clinically significant tumor control due to elements such as T cell localization, persistence, and function [90].

Across melanoma vaccine trials, vaccine site analyses have yielded actionable insights, including the impact of adjuvant, antigen, and injection strategy on immune cell infiltration, formation of TLS, antigen processing, and T cell homing [45,49,73]. Thus, the key challenge is to harness the potential of vaccine site analysis without overinterpreting its predictive power. The above gp100 findings demonstrate that local responses must be contextualized with systemic immune responses and clinical endpoint examination. Furthermore, this study highlights the potential effects of adjuvant formulation and delivery strategy on T cell differentiation, survival, and efficacy.

Although many of the previously discussed studies of vaccine site biology have been performed in melanoma patients, there is increasing interest in expanding these insights to other malignancies. In a 2025 phase I clinical trial, Braun and colleagues reported the significant development of T cell responses to vaccine antigens in all participants with high-risk, fully resected clear cell renal cell carcinoma after receiving a personalized neoantigen vaccine [91]. This study demonstrated that the VSME enabled durable peripheral T cell clonal expansion and reactivity against autologous tumors, suggesting successful priming and activation at the site of injection. As previously discussed, the 2013 study by Ohlfest and colleagues demonstrated the importance of vaccine site selection in immunotherapy trial design in a murine model of orthotopic intracranial glioma. Vaccination closer to the location of the tumor resulted in fewer antigen-specific CD8^+^ T cells, with lower TCR affinity and diminished effector function. This effect was most pronounced with vaccination in the neck, and diminished with vaccination in the hind leg, farther from the tumor of interest. This suggests that the local immunosuppressive environment created by the tumor compromised vaccine-induced T cell priming at the site [87]. A 2013 study by Barr and colleagues examined T cells at vaccine sites in a murine model of neuroblastoma, demonstrating that although T cell migration and proliferation at the vaccine site were similar between tumor-free and tumor-bearing mice, T cells at the vaccine site of tumor-bearing hosts were more apoptotic and produced lower levels of inflammatory cytokines. The results of this study suggested that tumor burden contributed to a systemic immunosuppressive state and widespread immune dysregulation driven by the tumor, which underscored a key factor in the development and previous failure of earlier cancer vaccines for neuroblastoma [92]. Both of these studies were conducted in murine models, and to our knowledge, similar analyses have not been conducted in humans. Although it is difficult to infer species specificity and translation to humans based on the existing literature, these changes in T cell quantity and function highlight the spatial influence of tumor-induced immunosuppression, which may be important to consider in vaccine strategy design. The central challenge in the widespread adoption of VSME analyses is derived from the marked differences in tumor antigen expression, tissue immune environments, and adjuvant responsiveness.

Interpretation of vaccine site findings is further complicated by the heterogeneity in sampling and analysis methods. As an example, excisional biopsies provide intact tissue architecture and allow for comprehensive histologic and spatial analyses, but they are invasive and often not available for repeated sampling [93]. Fine-needle aspiration (FNA) offers a less invasive alternative in melanoma, but it yields different cellular compositions, lower accuracy, and limited tissue architecture, and may undersample key compartments, such as stromal or follicular DC networks [93,94,95]. These considerations are summarized in Table 2. In animal models, most studies include harvest of the skin following euthanasia, allowing analysis of the full-thickness vaccine site and surrounding tissues [43,44,48,65]. In human studies, techniques are more variable, ranging from 3–4 mm punch biopsies [45,47,49,65,70,73] to full surgical excision of vaccine sites and surrounding tissues [45,46,47,73,84]. Additionally, some studies do not specify details regarding biopsy techniques [42,64]. Furthermore, significant findings regarding VSME mechanics have emerged from studies directly comparing different biopsy techniques within the same work; for example, analyses have collectively assessed patient samples derived from punch biopsies and surgical excision [45,47,73]. Even within a single study, differences in timing of sampling relative to vaccine administration have the potential to alter observed cell populations and cytokine profiles [49,96]. Collectively, these factors introduce variability and potential bias, which limit the direct comparability between studies and the generalizability of findings regarding vaccine-induced immune responses.

A central translational goal in VSME analysis is to determine whether local immune activity at the site of injection has the potential to serve as a reliable predictor of systemic immunity and durable clinical benefit. While certain studies have documented a correlation between increased local T cell activation and subsequent tumor regression [45,73], others have failed to replicate these associations [97]. These inconsistent findings likely reflect differences in study design, adjuvant selection, and sampling and analytic methods across vaccine trials, leading to a paucity of literature contributing to the formal assessment of the predictive value of vaccine site data. With more robust and standardized datasets, the relationship between immune activation at the vaccine site and subsequent clinical outcomes may be more clearly delineated. Early responses, or lack thereof, at the vaccine site could inform modifications to treatment strategies, including the addition of concurrent checkpoint blockade, incorporation of additional adjuvants, or adjustment of antigen selection within the vaccine formulation. Conversely, strong early responses to vaccination may support a more tailored approach, allowing thoughtful limitation of vaccine regimen duration while still achieving an effective systemic immune response.

## 9. Future Directions

To further elucidate the role of vaccine site analysis, there is a need for standardized protocols governing vaccine site sampling, processing, and analysis across pilot studies. This would allow reproducibility across trials, facilitate pooled analyses, and enable more accurate interpretation of these findings.

Advances in high-dimensional immunologic profiling and advanced computational methods offer unprecedented opportunities to continue to explore vaccine site biology. Single-cell RNA sequencing, combined with spatial transcriptomics, has the potential to further enable gene expression mapping, analysis of tumor heterogeneity, investigation of interactions between tumor cells and their microenvironment, including stromal and immune cell components, and assessment of therapeutic impacts on gene expression and cell interactions [98]. Multi-modal single-cell technologies also have the ability to allow for simultaneous profiling of transcriptomes, epigenomes, and proteomes, which allows for a more comprehensive understanding of vaccine efficacy via a more in-depth assessment of immune cell states and functional responses to vaccination [99,100,101].

While single-cell analyses may provide valuable insights, they inherently disrupt tissue architecture and cell–cell interactions, a challenge that can be mediated through the use of spatial assessment and transcriptomics, as these approaches preserve the spatial organization of cells within intact tissue segments [102]. Emerging spatial profiling technologies, including the NanoString^®^ CosMx™ Spatial Molecular Imaging system, have facilitated high-resolution mapping of immune cell subpopulation localization and their spatial interactions with heterogeneous tumor cell populations [103,104]. These techniques may overcome the above challenges by preserving the physical context of immune cells within the sampled vaccine site [105], which may prove valuable in the use of biopsy results to personalize vaccine and adjuvant regimens. Spatial analyses of activated immune cells, TLS formation, mRNA expression, and chemokine secretion at the vaccine site may provide valuable insight into an individual’s immune response to vaccination. Such architecture-preserving data may reveal immune triads [83], TLS formation, and other spatially organized immune features that may serve as potential prognostic indicators of vaccine efficacy. This level of resolution may help distinguish antigen-specific immune responses from adjuvant-induced inflammation and could inform refinements in antigen selection, peptide length, adjuvant formulation, and overall vaccine design. Moreover, the comprehensive aggregation of spatially resolved multi-omics data may enable the development of neural network models capable of predicting clinical outcomes and guiding the personalization of vaccine strategies. Collectively, these approaches promise to advance our understanding of the VSME and unlock new insights and pathways towards personalized vaccination and cancer treatment.

## 10. Conclusions

Peptide vaccines remain a promising experimental area within melanoma therapy, particularly when used in combination with immune checkpoint blockade or other adjuvant strategies. The biology of the VSME presents an underexplored potential resource for understanding local antigen presentation and the widespread immune responses that can result from targeted therapeutics. Clinical trials and targeted studies examining vaccine site biology have provided early insights into the quality and efficacy of local and systemic immune responses, offering critical information that is directly translatable to the design and optimization of cancer vaccines.

However, substantial challenges remain, including methodological heterogeneity of published studies, limited sample sizes, and the unclear predictive value of local responses. Addressing these obstacles will require future studies and coordinated efforts among the research community to standardize sampling, continue adoption of next-generation technologies, and incorporate vaccine site analysis into the design of future peptide vaccine trials. These steps may enable vaccine site assessment to serve as a predictive and actionable biomarker in the appraisal of cancer vaccines.

## Figures and Tables

**Figure 1 vaccines-13-01150-f001:**
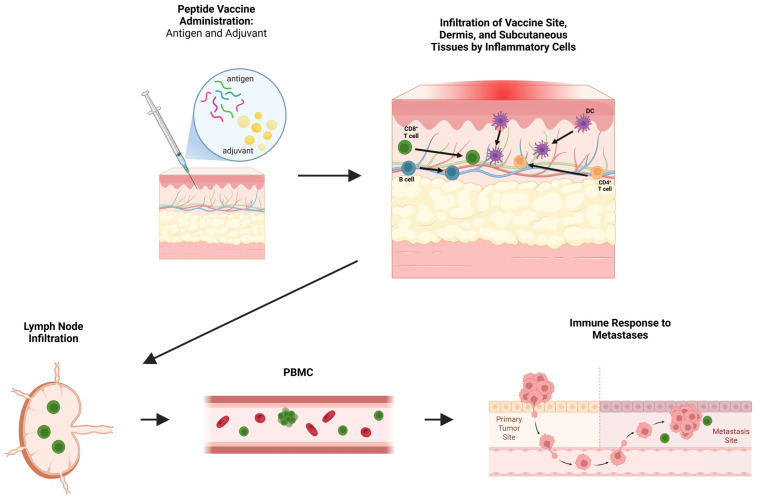
Overview of peptide vaccine-induced immune responses. Peptide vaccines and included adjuvants initiate local infiltration of inflammatory cells including B cells, dendritic cells (DCs), and CD4^+^ and CD8^+^ T cells into the vaccine site, dermis, and subcutaneous tissues. Antigen presentation and priming occur locally and within draining lymph nodes, leading to the expansion of effector populations detectable in peripheral blood mononuclear cells (PBMCs). These circulating antigen-specific immune cells may subsequently mediate recognition and immune responses against distant tumor sites. Figure created with BioRender (2025).

**Figure 2 vaccines-13-01150-f002:**
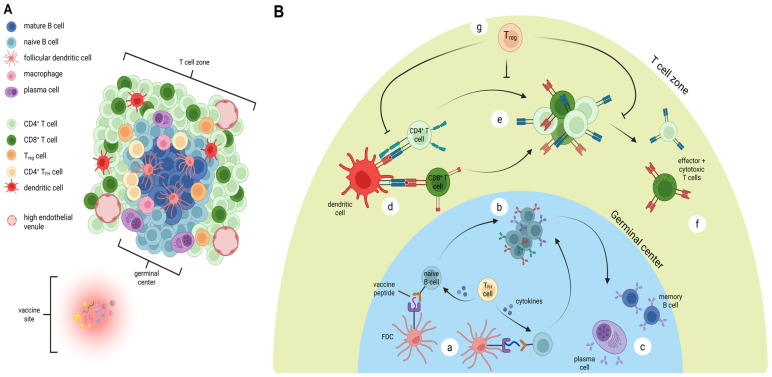
Organization and immune activity in vaccine-site-associated TLS. (**A**) Mature TLS formation near vaccination site, demonstrating germinal center composed of mature B cells, follicular dendritic cells, T regulatory (T_reg_) cells, macrophages, and CD4^+^ follicular helper T cells (T_FH_). The adjacent T cell zone contains CD4^+^ T cells, CD8^+^ T cells, mature dendritic cells, plasma cells, T_reg_ cells, and high endothelial venules. (**B**) Proposed critical mechanisms of antigen presentation within TLS. Follicular dendritic cells (FDC) present vaccine peptides to naïve B cells, with cytokine release from T_FH_ cells promoting activation (a). This leads to proliferation, class switching, and maturation of B cells (b), with ultimate generation of plasma cells and memory B cells (c). In the T cell zone, mature dendritic cells activate CD4^+^ and CD8^+^ T cells (d) leading to proliferation (e) and ultimately generation of CD4^+^ effector and CD8^+^ cytotoxic T cells (f). Each of these steps may be regulated by T_reg_ cells (g). Figure created with BioRender (2025).

**Table 2 vaccines-13-01150-t002:** Comparison of vaccine-site microenvironment (VSME) sampling methods with advantages and disadvantages.

VSME Sampling Method	Advantages	Disadvantages
Excisional or incisional biopsy	Preserve tissue architecture Allow comprehensive histologic and spatial analyses	Increased invasiveness May limit repeated sampling High variability in methods (punch biopsies vs. full surgical excision)
Fine needle aspiration (FNA)	Less invasive May allow repeated sampling over time	Do not accurately represent tissue architecture Limits histologic and spatial analyses Limited tissue available for analysis May undersample critical compartments and cell populations

## Data Availability

This manuscript does not present new data.

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
