# Peer review of "Shaping Antitumor Immunity with Peptide Vaccines: Implications of Immune Modulation at the Vaccine Site"

_vaccines, 2025, doi:10.3390/vaccines13111150_

Round 1

Reviewer 1 Report

Comments and Suggestions for Authors

In this review article, entitled “Shaping antitumor immunity with peptide vaccines: Implications of immune modulation at the vaccine site” by Amrita Sarkar et al., describes that, although cancer immunotherapies with peptide vaccines has been developed for cancers, evaluating peripheral immune reaction and clinical outcomes, analyzing the vaccine site microenvironment (VSME) may be important for the future vaccine development to predict vaccine-induced immune responses and effectiveness. The authors summarized recent findings regarding VSME biology with variety of vaccine strategies, including injection sites, adjuvants, timings, etc. They finally claims that the way to evaluate the VSME needs to be standardized for further deepened evaluations among many trials, leading to better interpretations of vaccine biology and vaccine development.  There are a couple of issues raised.

  1. In Figure 1, the total quality need to be improved. In left upper panel, the authors need to explain which is antigen and adjuvant. In right upper panel, the authors may indicate injection site as gradient red color. This red area should be located in the skin as injection site. In right lower panel, the letters in cartoon are too small for readers. Please enlarge those letters.
  2. The authors claims that standardization of VSME evaluation is necessary for better understanding for vaccine efficacy. It would be helpful if there is a table summering the variety ways of VSME evaluation in clinical trials.

Author Response

Response to Reviewer  Comments

Thank you very much for your comprehensive review of this manuscript. Our responses are detailed below. We appreciate the time and effort dedicated to review and feel the suggested revisions have helped to strengthen our paper.

Point-by-point response to Comments and Suggestions

Comment 1: In Figure 1, the total quality need to be improved. In left upper panel, the authors need to explain which is antigen and adjuvant. In right upper panel, the authors may indicate injection site as gradient red color. This red area should be located in the skin as injection site. In right lower panel, the letters in cartoon are too small for readers. Please enlarge those letters.

Response 1:  Thank you for pointing this out. We agree with this feedback, and have incorporated these changes into Figure 1. These changes can be found in the Figure 1 file, and in the manuscript text on page 8, line 195.

Comment 2: The authors claims that standardization of VSME evaluation is necessary for better understanding for vaccine efficacy. It would be helpful if there is a table summering the variety ways of VSME evaluation in clinical trials.

Response 2: Thank you for this suggestion. We have added this as Table 2: Comparison of Established VSME Sampling Methods, found on page 14, lines 408-409.

Table 2: Comparison of Established VSME Sampling Methods

VSME Sampling Method

Advantages

Disadvantages

Excisional or incisional biopsy

·        Preserve tissue architecture

·        Allow comprehensive histologic and spatial analyses

·        Increased invasiveness

·        May limit repeated sampling

·        High variability in methods (punch biopsies vs. full surgical excision)

Fine needle aspiration (FNA)

·        Less invasive

·        May allow repeated sampling over time

·        Do not accurately represent tissue architecture

·        Limits histologic and spatial analyses

·        Limited tissue available for analysis

·        May undersample critical compartments and cell populations

Table 2: Comparison of vaccine-site microenvironment (VSME) sampling methods with advantages and disadvantages

Reviewer 2 Report

Comments and Suggestions for Authors

In the review titled “Shaping Antitumor Immunity with Peptide Vaccines: Implications of Immune Modulation at the Vaccine Site”, Sarkar et. al. provides a valuable synthesis of the current understanding of the vaccine site microenvironment (VSME) and its critical role in shaping antitumor immunity following peptide vaccination. The authors successfully highlight the VSME as an under-explored niche that bridges local immune activation and systemic outcomes. The manuscript is well-researched, logically structured, and addresses a topic of significant importance in the field of cancer immunotherapy. The emphasis on the practical and translational challenges of VSME analysis is particularly appreciated. While the review is strong in its current form, several areas could be enhanced to improve its impact, clarity, and utility for the research community. The following points are offered as constructive suggestions for further refinement.

Major Points:

  1. The review covers all essential topics, but certain themes, such as the dual role of Incomplete Freund's Adjuvant (IFA) in T cell sequestration versus enhanced immunogenicity, are discussed in multiple sections (e.g., Sections 3, 4, and 6). This leads to some repetition and a slightly fragmented narrative.
  2. The manuscript accurately presents conflicting data regarding adjuvants like IFA, which in some studies causes detrimental T cell sequestration and in others enhances the magnitude and durability of T cell responses. However; the analysis of these discrepancies remains somewhat descriptive, a more mechanistic discussion would be highly beneficial.
  3. While Figure 1 provides a useful basic overview, the complexity of the concepts discussed would benefit significantly from additional visual aids. For example, a comparative diagram illustrating how different adjuvants (e.g., IFA, TLR agonists, QS-21) sculpt distinct immune cell compositions and cytokine milieus within the VSME, and/or a model of Tertiary Lymphoid Structure (TLS) formation at the vaccine site, depicting the key cellular components (T cells, B cells, DCs, PNAd+ high endothelial venules) and its hypothesized role in promoting effective immunity. Finally, a summary table or flowchart comparing the advantages and disadvantages of different VSME sampling techniques (e.g., punch biopsy, surgical excision, fine-needle aspiration) to guide future study design would be beneficial.
  4. The potential of the VSME as a biomarker is mentioned, but the discussion on how to leverage it for clinical decision-making is somewhat generic. Discuss how real-time or serial analysis of the VSME could, in the future, inform personalized adjustments to a vaccination regimen, would also be beneficial.
  5. The mention of single-cell and spatial transcriptomics is appropriate but could be more detailed regarding their practical application and integration. For example, briefly elaborate on how these technologies (e.g. Multi-omics Integration, Spatial Transcriptomics and/or AI-Assisted Analysis) can be specifically applied to the VSME.

Overall, this is a thoughtful and informative review that fills an important niche in the cancer vaccine literature. The authors have done an excellent job in compiling the relevant evidence. Addressing the points above, particularly by providing deeper mechanistic insight, concrete methodological recommendations, and enhanced visual and conceptual frameworks, would transform a very good review into an essential and foundational resource for the field.

Author Response

Response to Reviewer  Comments

Thank you very much for your detailed review of our manuscript. Our responses to each comment are detailed below. We appreciate the time and effort dedicated to this review and to strengthening our manuscript.

Point-by-point response to Comments and Suggestions

Comment 1: The review covers all essential topics, but certain themes, such as the dual role of Incomplete Freund’s Adjuvant (IFA) in T cell sequestration versus enhanced immunogenicity, are discussed in multiple sections (e.g. Sections 3, 4, and 6). This leads to some repetition and a slightly fragmented narrative.

Response 1:  We appreciate this point, and have significantly revised and consolidated repetitive portions of Sections 3 and 4.  The majority of this consolidation can be found on page 8-10, lines 207-260.

“Some concerns have been raised in murine models that vaccination with a minimal epitope melanoma peptide in IFA led to local inflammatory changes that recruited tumor-specific CD8+ T cells to the vaccine site instead of to the tumor itself. Furthermore, antigen-specific T cells died in the VSME [48]. These findings have raised concerns about use of IFA with peptide vaccines; however, those murine studies identified this concern only with a 9-mer (minimal epitope) peptide, but this was not replicated when a longer peptide was used in the vaccine. In contrast, vaccine formulations that led to less antigen persistence at the site (CD40 antibody plus TLR agonist) shifted T cell localization toward tumors [48]. Additionally, investigation by Hailemichael et al on the impact of adjuvant IFA with concomitant anti-CTLA-4 therapy demonstrated sequestration and destruction of anti-CTLA-4 induced effector T cells that were specific for non-vaccine tumor antigens [65]. In humans, vaccination with peptides in IFA also leads to T cell accumulation at vaccine sites [45]. One proposed mechanism for T cell retention in inflamed sites involves upregulation of integrins a1b1 (VLA-1) and aEb7 (CD103) which are markers of tissue-resident memory T cells and enable adherence of T cells to collagen IV and E-cadherin in peripheral tissues; these have been identified at high frequency in CD8+ T cell infiltrating the VSME (after peptide + IFA vaccines) and in the melanoma tumor microenvironment [46,66,67] and may contribute to both the acquisition of a memory-like phenotype and enhanced lymphocytic trafficking and motility [68].

Despite concerns about T cell accumulation at vaccine sites, human clinical trials do support use of IFA as a vaccine adjuvant [69]. Clinical trials in humans have tested whether IFA depleted antigen-specific T cells in circulation when added to a minimal epitope peptide vaccine using peptide plus a TLR agonist. In two trials, addition of IFA to peptide vaccines has enhanced immunogenicity and T cell responses, measured by IFN-g ELISpot assay, compared to vaccine administration with adjuvant TLR agonists alone [70,71]. The Mel58 trial [NCT01585350] demonstrated that IFA enhanced the magnitude and durability of CD8+ T cell responses to melanoma peptides when used in conjunction with TLR agonists, rather than diminishing them [70].

Thus, human clinical trial data do not support the same concerns raised in the murine studies mentioned above. Despite the elegant design of those murine studies, they did not reflect the setting of human cancer vaccines – in those murine studies, T cells transgenic for the melanoma antigen were given adoptively, leading to dramatically greater frequencies of melanoma antigen-reactive T cells than found in melanoma patients. Also, the dose of IFA administered to the mouse, if translated to the human on a weight:weight basis, would be comparable to giving about 500 mL of IFA, rather than the typical dose of about 1 mL in most human clinical trials. Thus, the selective homing of T cells to the vaccine sites in that mouse model probably do not reflect the normal biology in human cancer patients.

To understand cellular and molecular effects of peptide vaccines and their adjuvants, our group has studied vaccine sites after 1 or more vaccinations with IFA or other adjuvants, and with or without melanoma peptide vaccine. We have identified a high proportion of activated, antigen-specific CD8+ T cells in the VSME, particularly when peptide was present [46]. There was evidence that those T cells in the VSME may be dysfunctional, likely explained by continuous antigen stimulation at the VSME [46]. On the other hand, analyses of human vaccine sites after use of IFA have revealed increased markers of mature dendritic cells; increased CD40 and CD40 ligand expression [47], enhanced Th1 signaling [47,49], and reduced arginase expression [47], especially after 3 vaccines at the same site. Collectively, current evidence indicates that IFA significantly shapes the VSME through inflammatory signaling, immune cell recruitment, and enhancement of downstream systemic immune responses relative to other adjuvants. Additional studies have further supported that adjuvants including TLR agonists (i.e. poly-ICLC and CpG) can support DC activation and T cell priming, either with TLR agonists alone or by adding TLR agonists to IFA [49,50,72]. However, many of the precise molecular and cellular mechanisms underlying IFA-induced alterations within the VSME remain incompletely understood, warranting further detailed investigation. “

Comment 2: The manuscript accurately presents conflicting data regarding adjuvants like IFA, which in some studies causes detrimental T cell sequestration and in others enhances the magnitude and durability of T cell responses. However; the analysis of these discrepancies remains somewhat descriptive, a more mechanistic discussion would be highly beneficial.  

Response 2: Thank you for this feedback. We have expanded our section addressing mechanisms of adjuvant action. This can be found on pages 4-5, lines 141-149, 151-168.

“Aluminum adjuvants have been widely utilized as vaccine adjuvants, in both humans and animals, as stimulators of the Th2 response for antibody responses [31]. These compounds form insoluble particles to stimulate DC [32] and macrophage uptake, resulting in inflammasome activation and production of pro-inflammatory cytokines [33]. A notable limitation of these adjuvants with regard to their use in cancer vaccines is their poor induction of Th1 response and cellular immunity. These limitations have been addressed through administration with additional adjuvants [34], or through bioengineering techniques including mesostrand [35] and nanoparticle [36] creation intended to augment Th1 responses.”

“Early evidence supporting this included significant increases in antibody responses to injected ovalbumin with addition of mineral-oil adjuvant. This increased response was noted to persist for over a year after injection, significantly longer than injection without adjuvant. Emulsions isolated from mice months after injection demonstrated a stratified appearance with an oil layer separated from the emulsion layer, supporting the hypothesis that prolonged breakdown of the emulsion allows depot formation with slow antigen release over time [38]. A study of early inflammatory changes following intradermal injection of incomplete Freund’s adjuvant (IFA), a water-in-oil emulsion, demonstrated significant increases of TNF-a and IFN-g mRNA in draining inguinal lymph nodes. With the addition of type II collagen to IFA, levels of TNF-a, IL-2, and IFN-g were increased; harvested lymph node cells also produced a strong TNF-a mRNA response following in vitro re-stimulation with type II collagen [39]. Examination of mouse splenocytes following intraperitoneal administration of IFA has demonstrated increases in IL-1, 4, 5, 6, and 13 mRNA-producing cells; these chemokine changes may contribute to initiation of the immune response, Th2 stimulation, class switching, B cell proliferation, and antibody production [40]. Additionally, inflammatory responses to oil-based adjuvants may involve generation of oxygen radicals, as they are attenuated by catalase and antioxidant administration [41]. “

Comment 3: While Figure 1 provides a useful basic overview, the complexity of the concepts discussed would benefit significantly from additional visual aids. For example, a comparative diagram illustrating how different adjuvants (e.g., IFA, TLR agonists, QS-21) sculpt distinct immune cell compositions and cytokine milieus within the VSME, and/or a model of Tertiary Lymphoid Structure (TLS) formation at the vaccine site, depicting the key cellular components (T cells, B cells, DCs, PNAd+ high endothelial venules) and its hypothesized role in promoting effective immunity. Finally, a summary table or flowchart comparing the advantages and disadvantages of different VSME sampling techniques (e.g., punch biopsy, surgical excision, fine-needle aspiration) to guide future study design would be beneficial.

Response 3: We have added a new figure (Figure 2), which can be seen on page 10, lines 262-263. Additionally, we have added a table (Table 2) comparing VSME sampling techniques on page 14, lines 406-407. This is also copied below.

Table 2: Comparison of Established VSME Sampling Methods

VSME Sampling Method

Advantages

Disadvantages

Excisional or incisional biopsy

·        Preserve tissue architecture

·        Allow comprehensive histologic and spatial analyses

·        Increased invasiveness

·        May limit repeated sampling

·        High variability in methods (punch biopsies vs. full surgical excision)

Fine needle aspiration (FNA)

·        Less invasive

·        May allow repeated sampling over time

·        Do not accurately represent tissue architecture

·        Limits histologic and spatial analyses

·        Limited tissue available for analysis

·        May undersample critical compartments and cell populations

Comment 4: The potential of the VSME as a biomarker is mentioned, but the discussion on how to leverage it for clinical decision-making is somewhat generic. Discuss how real-time or serial analysis of the VSME, could, in the future, inform personalized adjustments to a vaccination regimen, would also be beneficial.

Response 4: We have expanded on this in section 8, which can be found on page 14, lines 415-422.

“With more robust and standardized datasets, the relationship between immune activation at the vaccine site and subsequent clinical outcomes may be more clearly delineated. Early responses, or lack thereof, at the vaccine site could inform modifications to treatment strategies, including the addition of concurrent checkpoint blockade, incorporation of additional adjuvants, or adjustment of antigen selection within the vaccine formulation. Conversely, strong early responses to vaccination may support a more tailored approach, allowing thoughtful limitation of vaccine regimen duration while still achieving an effective systemic immune response.”

Comment 5: The mention of single-cell and spatial transcriptomics is appropriate but could be more detailed regarding their practical application and integration. For example, briefly elaborate on how these technologies (e.g. Multi-omics Integration, Spatial Transcriptomics and/or AI-Assisted Analysis) can be specifically applied to the VSME.

Response 5:  Thank you for this suggestion. We have elaborated on this on page 15, lines 441-444, and 447-459.

“Emerging spatial profiling technologies, including the NanoString® CosMx™ Spatial Molecular Imaging system, have facilitated high-resolution mapping of immune cell subpopulation localization and their spatial interactions with heterogeneous tumor cell populations [103,104].”

“Spatial analyses of activated immune cells, TLS formation, mRNA expression, and chemokine secretion at the vaccine site may provide valuable insight into an individual’s immune response to vaccination. Such architecture-preserving data may reveal immune triads [83], TLS formation, and other spatially organized immune features that may serve as potential prognostic indicators of vaccine efficacy. This level of resolution may help distinguish antigen-specific immune responses from adjuvant-induced inflammation and could inform refinements in antigen selection, peptide length, adjuvant formulation, and overall vaccine design. Moreover, the comprehensive aggregation of spatially resolved multi-omics data may enable the development of neural network models capable of predicting clinical outcomes and guiding the personalization of vaccine strategies. Collectively, these approaches promise to advance our understanding of the VSME and unlock new insights and pathways towards personalized vaccination and cancer treatment.”

Reviewer 3 Report

Comments and Suggestions for Authors

The manuscript entitled "Shaping Antitumor Immunity with Peptide Vaccines: Implications of Immune Modulation at the Vaccine Site" is a small review work written in clear understandable English. Authors focused the attention of the reader on the questions of selecting the site of vaccination of peptide vaccines. In general, the manuscript is well performed, however I have few Minor recommendations for the authors.

  1. In the chapter 3 it will be good to provide some comparative table of different adjuvants and their action, benefits and drawbacks.
  2. You may just a little bit enlarge Introduction section, provide here brief discussion-comparison of peptide vaccines and modern vaccines such as adenoviral vaccines, mRNA-vaccines (in context of anti-cancer vaccines).
  3. Lines 286-290. Mice is small. Will it be observed the same effect at human? Which is the area of the inhibiting immunization? Is this effect specie-specific?

Author Response

Response to Reviewer  Comments

Thank you very much for your thorough review of our manuscript. Our responses to each point are detailed below. We appreciate the time and effort dedicated to reviewing and strengthening our paper.

Point-by-point response to Comments and Suggestions

Comment 1: In the chapter 3 it will be good to provide some comparative table of different adjuvants and their action, benefits and drawbacks.

Response 1:  Thank you for this suggestion. We have included a new table (Table 1), which can be seen on pages 5-7, line 187, and below.

Table 1: Comparison of Adjuvants Utilized with Cancer Peptide Vaccines

Adjuvant

Mechanism

Advantages

Disadvantages

Mineral salts

·        Depot

·        Th2 activation

·        Safe, well-tolerated [31]

·        Novel engineering techniques allow improved Th1 activation [35, 36]

·        May be utilized with additional adjuvants to augment response [34]

·        Primarily Th2 activators with suboptimal activation of Th1 response and cellular immunity

Water-in-oil emulsions

·        Depot

·        Safe, generally well-tolerated

·        Augmentation of immunogenicity and T cell responses [45, 46]

·        Evidence of activated T cell recruitment to VSME [47]

·        Local adverse inflammatory effects

·        Evidence of T cell sequestration and dysfunction at vaccine site [48]

TLR agonists

·        Immunopotentiation

     CpG    oligodeoxynucleotides

·        Induce DC maturation with subsequent lymphocytic migration to VSME [42]

·        Limited antigen-specific adjuvant effect [42]

     Poly-ICLC

·        Enhanced DC activation, maturation, cross-presentation [49]

·        Increased infiltration of CD8+ T cells into tumor [50]

·        Transient effects on VSME cellular changes [49]

·        May have increased effectiveness with systemic administration [50]

     TLR2 agonist

·        Successful maturation of DCs and activation of APCs (51, 52)

·        Enhancement of Th1 activation (51)

·        Tumor-reactive T cell responses need further investigation (53)

     TLR1/2 agonist

·        Induction of CD8+ and Th1 responses in human (54)

·        Effector memory T cell accumulation at injection site (54)

·        Immune response demonstrated in one human subject (54)

·        Phase I trials in process (55)

Saponins

·        Immunopotentiation

     QS-21 (AS15)

·        Induction of CD8+ T cell responses [56]

·        Increased antibody production and cellular response after peptide vaccination [57]

·        Effective with intradermal administration [58]

·        Local inflammatory and grade 1-2 systemic adverse effects [57, 58]

·        No clinically significant effect compared to placebo in phase III trial [59]

IL-2/anti-IL-2 complex

·        Immunopotentiation

·        Drives DC expansion at vaccine site, draining lymph nodes, spleen [43]

·        Increases CD8+ T cell and NK cells at vaccine site [43]

·        Assessed in combination with mesoporous silica rod vaccine [43]

GM-CSF

·        Immunopotentiation

·        Generally safe and tolerated [60]

·        Some studies demonstrating improved immune response [61]

·        Some local inflammatory and systemic adverse effects [60]

·        Randomized trials show decreased T cell responses or no impact, when added to other effective adjuvants [60, 62, 63] 

Stabilizing gel matrices

·        Depot

·        Sustained cytokine release at vaccine site

·        Enhanced inflammatory infiltrate and antigen-specific immunity [44]

·        Early phase development [44]

Comment 2: You may just a little bit enlarge Introduction section, provide here brief discussion-comparison of peptide vaccines and modern vaccines such as adenoviral vaccines, mRNA-vaccines (in context of anti-cancer vaccines).

Response 2: We have expanded on this on page 2, lines 54-59, and lines 61-74.

“mRNA-based vaccines have been explored extensively as cancer therapeutics; they offer relative ease of protein expression compared to DNA vaccines and lower risk of incorporation into the host genome [6]. These vaccines do not contain an infectious component, and are typically well tolerated, with rapid degradation of vaccine components reducing risk of long-term adverse effects [7].”

“Early challenges in mRNA vaccine development included the inherent instability of these antigens, which have since been addressed through molecular engineering approaches designed to enhance stability, delivery, and uptake [6,7]. Although mRNA vaccines are attractive for their capacity to enable personalized vaccines based on individual tumor neoantigens, peptide-based vaccines may offer additional stability and implementation across diverse clinical settings [10]. Tumor cell-based vaccines can utilize whole tumor cells, often irradiated to reduce disease risk, to produce anti-tumor immune responses [11]. Alternatively, dendritic cell (DC) vaccines may use dendritic cells matured and stimulated in vitro to overcome tumor microenvironment immunosuppression, which may ordinarily limit DC function [11,12]. Viral vectors have also been explored as mechanisms to express tumor antigens or infect APCs to express tumor antigen transgenes. Although these vectors are often inherently immunogenic, vector-specific challenges must be considered in vaccine design, including host immunity to the viral vector and potential for neurotropism and infectious complications [13]. Vaccine types with lower inherent immunogenicity may require administration with local adjuvants to increase vaccine response and tumor infiltration by immunologically active cells [14].”

Comment 3: Lines 286-290. Mice is small. Will it be observed the same effect at human? Which is the area of the inhibiting immunization? Is this effect specie-specific?

Response 3: We have addressed this on page 13, lines 367-369, and lines 378-383.

“This effect was most pronounced with vaccination in the neck, and diminished with vaccination in the hind leg, farther from the tumor of interest.”

“Both of these studies were conducted in murine models, and to our knowledge, similar analyses have not been conducted in humans. Although it is difficult to infer species specificity and translation to humans based on the existing literature, these changes in T cell quantity and function highlight the spatial influence of tumor-induced immunosuppression, which may be important to consider in vaccine strategy design.”

Round 2

Reviewer 1 Report

Comments and Suggestions for Authors

In the revised manuscript, the authors addressed all questions reviewer asked.